# Online PCA for Contaminated Data

**Jiashi Feng**
ECE Department
National University of Singapore
jiashi@nus.edu.sg

**Huan Xu**
ME Department
National University of Singapore
mpexuh@nus.edu.sg

**Shie Mannor**
EE Department
Technion
shie@ee.technion.ac.il

**Shuicheng Yan**
ECE Department
National University of Singapore
eleyans@nus.edu.sg

## Abstract

We consider the online Principal Component Analysis (PCA) where *contaminated* samples (containing outliers) are revealed sequentially to the Principal Components (PCs) estimator. Due to their sensitiveness to outliers, previous online PCA algorithms fail in this case and their results can be arbitrarily skewed by the outliers. Here we propose the online robust PCA algorithm, which is able to improve the PCs estimation upon an initial one steadily, even when faced with a constant fraction of outliers. We show that the final result of the proposed online RPCA has an acceptable degradation from the optimum. Actually, under mild conditions, online RPCA achieves the maximal robustness with a $50\%$ breakdown point. Moreover, online RPCA is shown to be efficient for both storage and computation, since it need not re-explore the previous samples as in traditional robust PCA algorithms. This endows online RPCA with scalability for large scale data.

## 1 Introduction

In this paper, we investigate the problem of robust **P**rincipal **C**omponent **A**nalysis (PCA) in an online fashion. PCA aims to construct a low-dimensional subspace based on a set of principal components (PCs) to approximate all the observed samples in the least-square sense [19]. Conventionally, it computes PCs as the eigenvectors of the sample covariance matrix in *batch mode*, which is both computationally expensive and in particular memory exhausting, when dealing with large scale data. To address this problem, several online PCA algorithms have been developed in literature [15, 23, 10]. For online PCA, at each time instance, a new sample is revealed, and the PCs estimation is updated accordingly without having to re-explore all previous samples. Significant advantages of online PCA algorithms include independence of their storage space requirement of the number of samples, and handling newly revealed samples quite efficiently.

Due to the quadratic loss used, PCA is notoriously sensitive to corrupted observations (outliers), and the quality of its output can suffer severely in the face of even a few outliers. Therefore, much work has been dedicated to robustifying PCA [12, 2, 24, 6]. However, all of these methods work in batch mode and cannot handle sequentially revealed samples in the online learning framework. For instance, [24] proposed a high-dimensional robust PCA (HR-PCA) algorithm that is based on iterative performing PCA and randomized removal. Notice that the random removal process involves calculating the order statistics over all the samples to obtain the removal probability. Therefore, all samples must be stored in memory throughout the process. This hinders its application to large scale data, for which storing all data is impractical.

In this work, we propose a novel online Robust PCA algorithm to handle contaminated sample set, i.e., sample set that comprises both authentic samples (non-corrupted samples) and outliers (corrupted samples), which are revealed sequentially to the algorithm. Previous online PCA algorithms generally fail in this case, since they update the PCs estimation through minimizing the quadratic error w.r.t. every new sample and are thus sensitive to outliers. The outliers may manipulate the PCs estimation severely and the result can be arbitrarily bad. In contrast, the proposed online RPCA is shown to be robust to the outliers. This is achieved by a probabilistic admiting/rejection procedure when a new sample comes. This is different from previous online PCA methods, where each and every new sample is admitted. The probabilistic admittion/rejection procedure endows online RPCA with the ability to reject more outliers than authentic samples and thus alleviates the affect of outliers and robustifies the PCs estimation. Indeed, we show that given a proper initial estimation, online RPCA is able to steadily improve its output until convergence. We further bound the deviation of the final output from the optimal solution. In fact, under mild conditions, online RPCA can be resistent to $50\%$ outliers, namely having a $50\%$ breakdown point. This is the maximal robustness that can be achieved by any method.

Compared with previous robust PCA methods (typically works in batch mode), online RPCA only needs to maintain a covariance matrix whose size is independent of the number of data points. Upon accepting a newly revealed sample, online RPCA updates the PCs estimation accordingly without re-exploring the previous samples. Thus, online RPCA can deal with large amounts of data with low *storage expense*. This is in stark contrast with previous robust PCA methods which typically requires to remember all samples. To the best of our knowledge, this is the first attempt to make online PCA work for outlier-corrupted data, with theoretical performance guarantees.

## 2   Related Work

Standard PCA is performed in batch mode, and its high computational complexity may become cumbersome for the large datasets. To address this issue, different online learning techniques have been proposed, for example [1, 8], and many others.

Most of current online PCA methods perform the PCs estimation in an incremental manner [8, 18, 25]. They maintain a covariance matrix or current PCs estimation, and update it according to the new sample incrementally. Those methods provide similar PCs estimation accuracy. Recently, a randomized online PCA algorithm was proposed by [23], whose objective is to minimize the total expected quadratic error minus the total error of the batch algorithm (*i.e.*, the regret). However, none of these online PCA algorithms is robust to the outliers.

To overcome the sensitiveness of PCA to outliers, many robust PCA algorithms have been proposed [21, 4, 12], which can be roughly categorized into two groups. They either pursue robust estimation of the covariance matrix, *e.g.*, $M$-estimator [17], $S$-estimator [22], and Minimum Covariance Determinant (MCD) estimator [21], or directly maximize certain robust estimation of univariate variance for the projected observations [14, 3, 4, 13]. These algorithms inherit the robustness characteristics of the adopted estimators and are qualitatively robust. However, none of them can be directly applied in online learning setting. Recently, [24] and the following work [6] propose high-dimensional robust PCA, which can achieve maximum $50\%$ breakdown point. However, these methods iteratively remove the observations or tunes the observations weights based on statistics obtained from the whole data set. Thus, when a new data point is revealed, these methods need to re-explore all of the data and become quite expensive in computation and in storage.

The most related works to ours are the following two works. In [15], an incremental and robust subspace learning method is proposed. The method proposes to integrate the $M$-estimation into the standard incremental PCA calculation. Specifically, each newly coming data point is re-weighted by a pre-defined influence function [11] of its residual to the current estimated subspace. However, no performance guarantee is provided in this work. Moreover, the performance of the proposed algorithm relies on the accuracy of PCs obtained previously. And the error will be cumulated inevitably. Recently, a compressive sensing based recursive robust PCA algorithm was proposed in [20]. In this work, the authors focused on the case where the *outliers* can be modeled as *sparse vectors*. In contrast, we do not impose any structural assumption on the outliers. Moreover, the proposed method in [20] essentially solves compressive sensing optimization over a small batch of data to update the PCs estimation instead of using a single sample, and it is not clear how to extend the method to the

latter case. Recently, He *et al.* propose an incremental gradient descent method on Grassmannian manifold for solving the robust PCA problem, named GRASTA [9]. However, they also focus on a different case from ours where the outliers are sparse vectors.

## 3 The Algorithm

### 3.1 Problem Setup

Given a set of observations $\{\mathbf{y}_1, \cdots, \mathbf{y}_T\}$ (here $T$ can be finite or infinite) which are revealed sequentially, the goal of online PCA is to estimate and update the principal components (PCs) based on the newly revealed sample $\mathbf{y}_t$ at time instance $t$. Here, the observations are the mixture of authentic samples (non-corrupted samples) and outliers (corrupted samples). The authentic samples $\mathbf{z}_i \in \mathbb{R}^p$ are generated through a linear mapping: $\mathbf{z}_i = A\mathbf{x}_i + \mathbf{n}_i$. Noise $\mathbf{n}_i$ is sampled from normal distribution $\mathcal{N}(\mathbf{0}, I_p)$; and the signal $\mathbf{x}_i \in \mathbb{R}^d$ are i.i.d. samples of a random variable $\mathbf{x}$ with mean zero and variance $I_d$. Let $\mu$ denote the distribution of $\mathbf{x}$. The matrix $A \in \mathbb{R}^{p \times d}$ and the distribution $\mu$ are unknown. We assume $\mu$ is absolutely continuous w.r.t. the Borel measure and spherically symmetric. And $\mu$ has light tails, *i.e.*, there exist constants $C > 0$ such that $\Pr(\|\mathbf{x}\| \geq x) \leq d \exp(1 - Cx/\alpha\sqrt{d})$ for all $x \geq 0$. The outliers are denoted as $\mathbf{o}_i \in \mathbb{R}^p$ and in particular they are defined as follows.

**Definition 1** (Outlier). *A sample $\mathbf{o}_i \in \mathbb{R}^p$ is an outlier w.r.t. the subspace spanned by $\{\overline{\mathbf{w}}_j\}_{j=1}^d$ if it deviates from the subspace,* i.e., $\sum_{j=1}^d |\overline{\mathbf{w}}_j^T \mathbf{o}_i|^2 \leq \Gamma_o$.

In the above definition, we assume that the basis $\overline{\mathbf{w}}_j$ and outliers $\mathbf{o}$ are both *normalized* (see Algorithm 1 step 1)-a) where all the samples are $\ell_2$-normalized). Thus, we directly use inner product to define $\Gamma_o$. Namely a sample is called outlier if it is distant from the underlying subspace of the signal. Apart from this assumption, the outliers are arbitrary. In this work, we are interested in the case where the outliers are mixed with authentic samples uniformly in the data stream, *i.e.*, taking any subset of the dataset, the outlier fraction is identical when the size of the subset is large enough.

The input to the proposed online RPCA algorithm is the sequence of observations $\mathcal{Y} = \{\mathbf{y}_1, \mathbf{y}_2, \cdots, \mathbf{y}_T\}$, which is the union of authentic samples $\mathcal{Z} = \{\mathbf{z}_i\}$ generated by the aforementioned linear model and outliers $\mathcal{O} = \{\mathbf{o}_i\}$. The outlier fraction in the observations is denoted as $\lambda$. Online RPCA aims at learning the PCs robustly and the learning process proceeds in time instances. At the time instance $t$, online RPCA chooses a set of principal components $\{\mathbf{w}_j^{(t)}\}_{j=1}^d$. The performance of the estimation is measured by the Expressed Variance (E.V.) [24]:

$$\text{E.V.} \triangleq \frac{\sum_{j=1}^d \mathbf{w}_j^{(t)^T} AA^T \mathbf{w}_j^{(t)}}{\sum_{j=1}^d \overline{\mathbf{w}}_j^T AA^T \overline{\mathbf{w}}_j}.$$

Here, $\overline{\mathbf{w}}_j$ denotes the true principal components of matrix $A$. The E.V. represents the portion of signal $A\mathbf{x}$ being expressed by $\{\mathbf{w}_j^{(t)}\}_{j=1}^d$. Thus, $1 - \text{E.V.}$ is the reconstruction error of the signal. The E.V. is a commonly used evaluation metric for the PCA algorithms [24, 5]. It is always less than or equal to one, with equality achieved by a perfect recovery.

### 3.2 Online Robust PCA Algorithm

The details of the proposed online RPCA algorithm are shown in Algorithm 1. In the algorithm, the observation sequence $\mathcal{Y} = \{\mathbf{y}_1, \mathbf{y}_2, \cdots, \mathbf{y}_T\}$ is sequentially partitioned into $(T' + 1)$ batches $\{B_0, B_1, B_2, \ldots, B_{T'}\}$. Each batch consists of $b$ observations. Since the authentic samples and outliers are mixed uniformly, the outlier fraction in each batch is also $\lambda$. Namely, in each batch $B_i$, there are $(1 - \lambda)b$ authentic samples and $\lambda b$ outliers.

Note that such small batch partition is only for the ease of illustration and analysis. Since the algorithm only involves standard PCA computation, we can employ any incremental or online PCA method [8, 15] to update the PCs estimation upon accepting a new sample. The maintained sample covariance matrix, can be set to zero every $b$ time instances. Thus the batch partition is by no means necessary in practical implementation. In the algorithm, the initial PC estimation can be obtained through standard PCA or robust PCA [24] on a mini batch of the samples.

---
**Algorithm 1** Online Robust PCA Algorithm
---
**Input:** Data sequence $\{\mathbf{y}_1, \ldots, \mathbf{y}_T\}$, buffer size $b$.

**Initialization:** Partition the data sequence into small batches $\{B_0, B_1, \ldots, B_{T'}\}$. Each patch contains $b$ data points. Perform PCA on the first batch $B_0$ and obtain the initial principal component $\{\mathbf{w}_j^{(0)}\}_{j=1}^d$.

$t = 1$. $\mathbf{w}_j^* = \mathbf{w}_j^{(0)}, \forall j = 1, \ldots, d$.

**while** $t \leq T'$ **do**

   1) Initialize the sample covariance matrix: $C^{(t)} = 0$.

   **for** $i = 1$ to $b$ **do**

      a) Normalize the data point by its $\ell_2$-norm: $\mathbf{y}_i^{(t)} := \mathbf{y}_i^{(t)}/\|\mathbf{y}_i^{(t)}\|_{\ell_2}$.

      b) Calculate the variance of $\mathbf{y}_i^{(t)}$ along the direction $\mathbf{w}^{(t-1)}$: $\delta_i = \sum_{j=1}^d \left|\mathbf{w}_j^{(t-1)^T}\mathbf{y}_i^{(t)}\right|^2$.

      c) Accept $\mathbf{y}_i^{(t)}$ with probability $\delta_i$.

      d) Scale $\mathbf{y}_i^{(t)}$ as $\mathbf{y}_i^{(t)} \leftarrow \mathbf{y}_i^{(t)}/b\sqrt{\delta_i}$.

      e) If $\mathbf{y}_i^{(t)}$ is accepted, update $C^{(t)} \leftarrow C^{(t)} + \mathbf{y}_i^{(t)}\mathbf{y}_i^{(t)^T}$.

   **end for**

   2) Perform eigen-decomposition on $C_t$ and obtain the leading $d$ eigenvector $\{\mathbf{w}_j^{(t)}\}_{j=1}^d$.

   3) Update the PC as $\mathbf{w}_j^* = \mathbf{w}_j^{(t)}, \forall j = 1, \ldots, d$.

   4) $t := t + 1$.

**end while**

**Return** $\mathbf{w}^*$.

---

We now explain the intuition of the proposed online RPCA algorithm. Given an initial solution $\mathbf{w}^{(0)}$ which is "closer" to the true PC directions than to the outlier direction [1], the authentic samples will have larger variance along the current PC direction than outliers. Thus in the probabilistic data selection process (as shown in Algorithm 1 step b) to step d)), authentic samples are more likely to be accepted than outliers. Here the step d) of scaling the samples is important for obtaining an unbiased estimator (see details in the proof of Lemma 4 in supplementary material and [16]). Therefore, in the following PC updating based on standard PCA on the accepted data, authentic samples will contribute more than the outliers. The estimated PCs will be "moved" towards to the true PCs gradually. Such process is repeated until convergence.

## 4 Main Results

In this section we present the theoretical performance guarantee of the proposed online RPCA algorithm (Algorithm 1). In the sequel, $\mathbf{w}_j^{(t)}$ is the solution at the $t$-th time instance. Here without loss of generality we assume the matrix $A$ is normalized, such that the E.V. of the true principal component $\overline{\mathbf{w}}_j$ is $\sum_{j=1}^d \overline{\mathbf{w}}_j^T A^T A \overline{\mathbf{w}}_j = 1$. The following theorem provides the performance guarantee of Algorithm 1 under the noisy case. The performance of $\mathbf{w}^{(t)}$ can be measured by $H(\mathbf{w}^{(t)}) \triangleq \sum_{j=1}^d \|\mathbf{w}_j^{(t)^T}A\|^2$. Let $s = \|\mathbf{x}\|_2/\|\mathbf{n}\|_2$ be the *signal noise ratio*.

**Theorem 1** (Noisy Case Performance). *There exist constants $c_1', c_2'$ which depend on the signal noise ratio $s$ and $\epsilon_1, \epsilon_2 > 0$ which approximate zero when $s \to \infty$ or $b \to \infty$, such that if the initial solution $\mathbf{w}_j^{(0)}$ in Algorithm 1 satisfies:*

$$\sum_{i=1}^{\lambda b}\sum_{j=1}^d \left|\mathbf{w}_j^{(0)^T}\mathbf{o}_i\right|^2 \leq \frac{(1-\lambda)b(1-\epsilon^2)}{c_2'(1-\Gamma_o)}\left(\frac{1}{4}(c_1'(1-\epsilon)-\epsilon_1)^2 - \epsilon_2\right),$$

*and*

$$H(\mathbf{w}^{(0)}) \geq \frac{1}{2}(c_1'(1-2\epsilon)-\epsilon_1) - \sqrt{\frac{(c_1'(1-\epsilon)+\epsilon_1)^2 - 4\epsilon_2}{4} - \frac{c_2'\sum_{i=1}^{\lambda b}\sum_{j=1}^d (\mathbf{w}_j^{(0)^T}\mathbf{o}_i)^2(1-\Gamma_o)}{(1-\lambda)b(1-\epsilon^2)}},$$

*then the performance of the solution from Algorithm 1 will be improved in each iteration, and eventually converges to:*

$$\lim_{t\to\infty} H(\mathbf{w}^{(t)})$$

$$\geq \frac{1}{2}(c_1'(1-2\epsilon) - \epsilon_1) + \sqrt{\frac{(c_1'(1-2\epsilon) - \epsilon_1)^2 - 4\epsilon_2}{4} - \frac{c_2' \sum_{i=1}^{\lambda b} \sum_{j=1}^{d} (\mathbf{w}_j^{(0)^T} \mathbf{o}_i)^2 (1 - \Gamma_o)}{(1-\lambda)b(1-\epsilon^2)}}.$$

*Here $\epsilon_1$ and $\epsilon_2$ decay as $\tilde{O}(d^{\frac{1}{2}} b^{-\frac{1}{2}} s^{-1})$, $\epsilon$ decays as $\tilde{O}(d^{\frac{1}{2}} b^{-\frac{1}{2}})$, and $c_1' = (s-1)^2/(s+1)^2, c_2' = (1+1/s)^4$.*

**Remark 1.** From Theorem 1, we can observe followings:

1. When the outliers vanish, the second term in the square root of performance $H(\mathbf{w}^{(t)})$ is zero. $H(\mathbf{w}^{(t)})$ will converge to $(c_1'(1-2\epsilon) - \epsilon_1)/2 + \sqrt{(c_1'(1-2\epsilon) - \epsilon_1)^2 - 4\epsilon_2}/2 < c_1'(1-2\epsilon) - \epsilon_1 < c_1' < 1$. Namely, the final performance is smaller than but approximates 1. Here $c_1', \epsilon_1, \epsilon_2$ explain the affect of noise.

2. When $s \to \infty$, the affect of noise is eliminated, $\epsilon_1, \epsilon_2 \to 0, c_1' \to 1$. $H(\mathbf{w}^{(t)})$ converges to $1 - 2\epsilon$. Here $\epsilon$ depends on the ratio of intrinsic dimension over the sample size, and $\epsilon$ accounts for the statistical bias due to performing PCA on a small portion of the data.

3. When the batch size increases to infinity, $\epsilon \to 0$, $H(\mathbf{w}^{(t)})$ converges to 1, meaning perfect recovery.

To further investigate the behavior of the proposed online RPCA in presence of outliers, we consider the following noiseless case. For the noiseless case, the signal noise ratio $s \to \infty$, and thus $c_1', c_2' \to 1$ and $\epsilon_1, \epsilon_2 \to 0$. Then we can immediately obtain the performance bound of Algorithm 1 for the noiseless case from Theorem 1.

**Theorem 2** (Noiseless Case Performance). *Suppose there is no noise. If the initial solution $\mathbf{w}^{(0)}$ in Algorithm 1 satisfies:*

$$\sum_{i=1}^{\lambda b} \sum_{j=1}^{d} (\mathbf{w}_j^{(0)^T} \mathbf{o}_i)^2 \leq \frac{(1-\lambda)b}{4(1-\Gamma_o)},$$

*and*

$$H(\mathbf{w}^{(0)}) \geq \frac{1}{2} - \sqrt{\frac{1}{4} - \frac{\sum_{i=1}^{\lambda b} \sum_{j=1}^{d} (\mathbf{w}_j^{(0)^T} \mathbf{o}_i)^2 (1 - \Gamma_o)}{(1-\lambda)b}},$$

*then the performance of the solution from Algorithm 1 will be improved in each updating and eventually converges to:*

$$\lim_{t\to\infty} H(\mathbf{w}^{(t)}) \geq \frac{1}{2} + \sqrt{\frac{1}{4} - \frac{\sum_{i=1}^{\lambda b} \sum_{j=1}^{d} (\mathbf{w}_j^{(0)^T} \mathbf{o}_i)^2 (1 - \Gamma_o)}{(1-\lambda)b}}.$$

**Remark 2.** Observe from Theorem 2 the followings:

1. When the outliers are distributed on the groundtruth subspace, *i.e.*, $\sum_{j=1}^{d} |\overline{\mathbf{w}}_j^T \mathbf{o}_i|^2 = 1$, the conditions become $\sum_{i=1}^{\lambda b} \sum_{j=1}^{d} (\mathbf{w}^{(0)^T} \mathbf{o}_i)^2 < \infty$ and $H(\mathbf{w}^{(0)}) \geq 0$. Namely, for whatever initial solution, the final performance will converge to 1.

2. When the outliers are orthogonal to the groundtruth subspace, *i.e.*, $\sum_{j=1}^{d} |\overline{\mathbf{w}}_j^T \mathbf{o}_i|^2 = 0$, the conditions for the initial solution becomes $\sum_{i=1}^{\lambda b} \sum_{j=1}^{d} |\mathbf{w}_j^{(0)^T} \mathbf{o}_i|^2 \leq b(1-\lambda)/4$, and $H_0 \geq 1/2 - \sqrt{1/4 - \sum_{i=1}^{\lambda b} \sum_{j=1}^{d} (\mathbf{w}_j^{(0)^T} \mathbf{o}_i)^2/(1-\lambda)b}$. Hence, when the outlier fraction $\lambda$ increases, the initial solution should be further away from outliers.

3. When $0 < \sum_{j=1}^{d} |\overline{\mathbf{w}}_j^T \mathbf{o}_i|^2 < 1$, the performance of online RPCA is improved by at least $2\sqrt{1/4 - \sum_{i=1}^{\lambda b} \sum_{j=1}^{d} (\mathbf{w}_j^{(0)T} \mathbf{o}_i)^2 (1 - \Gamma_o)/(1 - \lambda)b}$ from its initial solution. Hence, when the initial solution is further away from the outliers, the outlier fraction is smaller, or the outliers are closer to groundtruth subspace, the improvement is more significant. Moreover, observe that given a proper initial solution, even if $\lambda = 0.5$, the performance of online RPCA still has a positive lower bound. Therefore, the breakdown point of online RPCA is $50\%$, the highest that any algorithm can achieve.

**Discussion on the initial condition**    In Theorem 1 and Theorem 2, a mild condition is imposed on the initial estimate. In practice, the initial estimate can be obtained by applying batch RPCA [6] or HRPCA [24] on a small subset of the data. These batch methods are able to provide initial estimate with performance guarantee, which may satisfy the initial condition.

## 5    Proof of The Results

We briefly explain the proof of Theorem 1: we first show that when the PCs estimation is being improved, the variance of outliers along the PCs will keep decreasing. Then we demonstrate that each PCs updating conducted by Algorithm 1 produces a better PCs estimation and decreases the impact of outliers. Such improvement will continue until convergence, and the final performance has bounded deviation from the optimum.

We provide here some concentration lemmas which are used in the proof of Theorem 1. The proof of these lemmas is provided in the supplementary material. We first show that with high probability, both the largest and smallest eigenvalues of the signals $\mathbf{x}_i$ in the original space converge to $1$. This result is adopted from [24].

**Lemma 1.** *There exists a constant c that only depends on $\mu$ and d, such that for all $\gamma > 0$ and b signals $\{\mathbf{x}_i\}_{i=1}^{b}$, the following holds with high probability:*

$$\sup_{\mathbf{w} \in \mathcal{S}_d} \left| \frac{1}{b} \sum_{i=1}^{b} (\mathbf{w}^T \mathbf{x}_i)^2 - 1 \right| \leq \epsilon,$$

*where $\epsilon = c\alpha \sqrt{d \log^3 b / b}$.*

Next lemma is about the sampling process in the Algorithm 1 from step b) to step d). Though the sampling process is without replacement and the sampled observations are not i.i.d., the following lemma provides the concentration of the sampled observations.

**Lemma 2** (Operator-Bernstein inequality [7])**.** *Let $\{\mathbf{z}_i'\}_{i=1}^{m}$ be a subset of $\mathcal{Z} = \{\mathbf{z}_i\}_{i=1}^{t}$, which is formed by randomly sampling* without *replacement from $\mathcal{Z}$, as in Algorithm 1. Then the following statement holds*

$$\left| \sum_{i=1}^{m} \mathbf{w}^T \mathbf{z}_i' - \mathbb{E}\left( \sum_{i=1}^{m} \mathbf{w}^T \mathbf{z}_i' \right) \right| \leq \delta$$

*with probability larger than $1 - 2\exp(-\delta^2/4m)$.*

Given the result in Lemma 1 , we obtain that the authentic samples concentration properties as stated in the following lemma [24].

**Lemma 3.** *If there exists $\epsilon$ such that*

$$\sup_{\mathbf{w} \in \mathcal{S}_d} \left| \frac{1}{t} \sum_{i=1}^{t} |\mathbf{w}^T \mathbf{x}_i|^2 - 1 \right| \leq \epsilon,$$

and the observations $\mathbf{z}_i$ are normalized by $\ell_2$-norm, then for any $\mathbf{w}_1, \cdots, \mathbf{w}_d \in \mathcal{S}_p$, the following holds:

$$\frac{(1-\epsilon)H(\mathbf{w}) - 2\sqrt{(1+\epsilon)H(\mathbf{w})}/s}{(1/s+1)^2}$$

$$\leq \quad \frac{1}{t}\sum_{i=1}^{t}\sum_{j=1}^{d}(\mathbf{w}_j^T\mathbf{z}_i)^2 \leq \frac{(1+\epsilon)H(\mathbf{w}) + 2\sqrt{(1+\epsilon)H(\mathbf{w})}/s + 1/s^2}{(1/s-1)^2},$$

where $H(\mathbf{w}) = \sum_{j=1}^{d}\|\mathbf{w}_j^T A\|^2$ and $s$ is the signal noise ratio.

Based on Lemma 2 and Lemma 3, we obtain the following concentration results for the selected observations in the Algorithm 1.

**Lemma 4.** *If there exists $\epsilon$ such that*

$$\sup_{\mathbf{w}\in\mathcal{S}_d}\left|\frac{1}{t}\sum_{i=1}^{t}|\mathbf{w}^T\mathbf{x}_i|^2 - 1\right| \leq \epsilon,$$

*and the observations $\{\mathbf{z}_i'\}_{i=1}^{m}$ are sampled from $\{\mathbf{z}_i\}_{i=1}^{d}$ as in Algorithm 1, then for any $\mathbf{w}_1, \ldots, \mathbf{w}_d \in \mathcal{S}_p$, with large probability, the following holds:*

$$\frac{(1-\epsilon)H(\mathbf{w}) - 2\sqrt{(1+\epsilon)H(\mathbf{w})}/s}{(1/s+1)^2 b/m} - \delta$$

$$\leq \quad \frac{1}{t}\sum_{i=1}^{t}\sum_{j=1}^{d}(\mathbf{w}_j^T\mathbf{z}_i')^2 \leq \frac{(1+\epsilon)H(\mathbf{w}) + 2\sqrt{(1+\epsilon)H(\mathbf{w})}/s + 1/s^2}{(1/s-1)^2 b/m} + \delta,$$

*where $H(\mathbf{w}) \triangleq \sum_{j=1}^{d}\|\mathbf{w}_j^T A\|^2$, $s$ is the signal noise ratio and $m$ is the number of sampled observations in each batch and $\delta > 0$ is a small constant.*

We denote the set of accepted authentic samples as $\mathcal{Z}_t$ and the set of accepted outliers as $\mathcal{O}_t$ from the $t$-th small batch. In the following lemma, we provide the estimation of number of accepted authentic samples $|\mathcal{Z}_t|$ and outliers $|\mathcal{O}_t|$.

**Lemma 5.** *For the current obtained principal components $\{\mathbf{w}_j^{(t-1)}\}_{j=1}^{d}$, the number of the accepted authentic samples $|\mathcal{Z}_t|$ and outliers $|\mathcal{O}_t|$ satisfy*

$$\left|\frac{|\mathcal{Z}_t|}{b} - \frac{1}{b}\sum_{i=1}^{(1-\lambda)b}\sum_{j=1}^{d}(\mathbf{w}_j^{(t-1)T}\mathbf{z}_i)^2\right| \leq \delta \text{ and } \left|\frac{|\mathcal{O}_t|}{b} - \frac{1}{b}\sum_{i=1}^{\lambda b}\sum_{j=1}^{d}(\mathbf{w}_j^{(t-1)T}\mathbf{o}_i)^2\right| \leq \delta$$

*with probability at least $1 - e^{-2\delta^2 b}$. Here $\delta > 0$ is a small constant, $\lambda$ is the outlier fraction and $b$ is the size of the small batch.*

From the above lemma, we can see that when the batch size $b$ is sufficiently large, the above estimation for $|\mathcal{Z}_t|$ and $|\mathcal{O}_t|$ holds with large probability. In the following lemma, we show that when the algorithm improves the PCs estimation, the impact of outliers will be decreased accordingly.

**Lemma 6.** *For an outlier $\mathbf{o}_i$, an arbitrary orthogonal basis $\{\mathbf{w}_j\}_{j=1}^{d}$ and the groundtruth basis $\{\overline{\mathbf{w}}_j\}_{j=1}^{d}$ which satisfy that $\sum_{j=1}^{d}\mathbf{w}_j^T\mathbf{o}_i \geq \sum_{j=1}^{d}\overline{\mathbf{w}}_j^T\mathbf{o}_i$ and $\sum_{j=1}^{d}\overline{\mathbf{w}}_j^T\mathbf{w}_j \geq \sum_{j=1}^{d}\overline{\mathbf{w}}_j^T\mathbf{o}_i$, the value of $\sum_{j=1}^{d}\mathbf{w}_j^T\mathbf{o}_i$ is a monotonically decreasing function of $\sum_{j=1}^{d}\overline{\mathbf{w}}_j^T\mathbf{w}_j$.*

Being equipped by the above lemmas, we can proceed to prove Theorem 1. The details of the proof is deferred to the supplementary material due to the space limit.

## 6 Simulations

The numerical study is aimed to illustrate the performance of online robust PCA algorithm. We follow the data generation method in [24] to randomly generate a $p \times d$ matrix $A$ and then scale its

leading singular value to $s$, which is the signal noise ratio. A $\lambda$ fraction of outliers are generated. Since it is hard to determine the most adversarial outlier distribution, in simulations, we generate the outliers concentrate on several directions deviating from the groundtruth subspace. This makes a rather adversarial case and is suitable for investigating the robustness of the proposed online RPCA algorithm. In the simulations, in total $T = 10,000$ samples are generated to form the sample sequence. For each parameter setting, we report the average result of 20 tests and standard deviation. The initial solution is obtained by performing batch HRPCA [24] on the first batch. Simulation results for $p = 100, d = 1, s = 2$ and outliers distributed on one direction are shown in Figure 1. It takes around $0.5$ seconds for the proposed online RPCA to process $10,000$ samples of $100$ dimensional, on a PC with Quad CPU with 2.83GHz and RAM of 8GB. In contrast, HRPCA costs 237 seconds to achieve E.V. $= 0.99$. More simulation results for the $d > 1$ case are provided in the supplementary material due to the space limit.

From the results, we can make the following observations. Firstly, online RPCA can improve the PC estimation steadily. With more samples being revealed, the E.V. of the online RPCA outputs keep increasing. Secondly, the performance of online RPCA is rather robust to outliers. For example, the final result converges to E.V. $\approx 0.95$ (HRPCA achieves $0.99$) even with $\lambda = 0.3$ for relatively low signal noise ratio $s = 2$ as shown in Figure 1. To more clearly demonstrate the robustness of online RPCA to outliers, we implement the online PCA proposed in [23] as baseline for the $\sigma = 2$ case. The results are presented in Figure 1, from which we can observe that the performance of online PCA drops due to the sensitiveness to newly coming outliers. When the outlier fraction $\lambda \geq 0.1$, the online PCA cannot recover the true PC directions and the performance is as low as $0$.

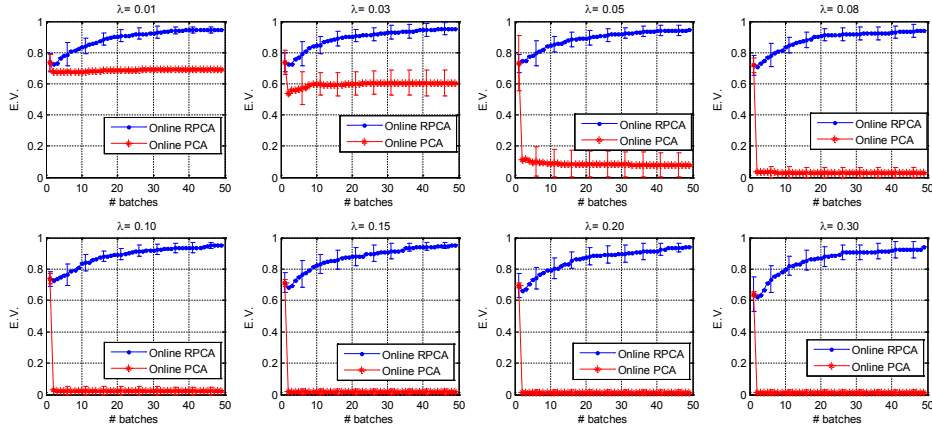

Figure 1: Performance comparison of online RPCA (blue line) with online PCA (red line). Here $s = 2, p = 100, T = 10,000, d = 1$. The outliers are distributed on a single direction.

## 7 Conclusions

In this work, we proposed an online robust PCA (online RPCA) algorithm for samples corrupted by outliers. The online RPCA alternates between standard PCA for updating PCs and probabilistic selection of the new samples which alleviates the impact of outliers. Theoretical analysis showed that the online RPCA could improve the PC estimation steadily and provided results with bounded deviation from the optimum. To the best of our knowledge, this is the first work to investigate such online robust PCA problem with theoretical performance guarantee. The proposed online robust PCA algorithm can be applied to handle challenges imposed by the modern big data analysis.

**Acknowledgement**

J. Feng and S. Yan are supported by the Singapore National Research Foundation under its International Research Centre @Singapore Funding Initiative and administered by the IDM Programme Office. H. Xu is partially supported by the Ministry of Education of Singapore through AcRF Tier Two grant R-265-000-443-112 and NUS startup grant R-265-000-384-133. S. Mannor is partially supported by the Israel Science Foundation (under grant 920/12) and by the Intel Collaborative Research Institute for Computational Intelligence (ICRI-CI).

## Footnotes

[1]In the following section, we will provide a precise description of the required closeness.

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
