[Supplementary Material]

# Online PCA for Contaminated Data
## Supplementary Material

**Jiashi Feng**
ECE Department
National University of Singapore
jiashi@nus.edu.sg

**Huan Xu**
ME Department
National University of Singapore
mpexuh@nus.edu.sg

**Shie Mannor**
EE Department
Technion
shie@ee.technion.ac.il

**Shuicheng Yan**
ECE Department
National University of Singapore
eleyans@nus.edu.sg

## 1   Technical Lemmas

Before proving the theoretical results in this paper, we first present following lemmas used in the proof.

**Lemma 1.** *There exists a constant $c$ that only depends on $\mu$ and $d$, such that for all $\gamma > 0$ and $b$ signals $\{\mathbf{x}_i\}_{i=1}^b$, the following holds with high probability:*

$$\sup_{\mathbf{w} \in \mathcal{S}_d} \left| \frac{1}{b} \sum_{i=1}^b (\mathbf{w}^T \mathbf{x}_i)^2 - 1 \right| \le \epsilon,$$

*where $\epsilon = c\alpha \sqrt{d \log^3 b / b}$.*

**Lemma 2** (Operator-Bernstein inequality). *Let $\{\mathbf{z}_i'\}_{i=1}^m$ be a subset of $\mathcal{Z} = \{\mathbf{z}_i\}_{i=1}^t$, which is formed by randomly sampling without replacement from $\mathcal{Z}$, as in Algorithm 1. Then the following statement holds*

$$\left| \sum_{i=1}^m \mathbf{w}^T \mathbf{z}_i' - \mathbb{E}\left( \sum_{i=1}^m \mathbf{w}^T \mathbf{z}_i' \right) \right| \le \delta$$

*with probability larger than $1 - 2\exp(-\delta^2/4m)$.*

## 2   Proof of Lemma 3

**Lemma 3.** *If there exists $\epsilon$ such that*

$$\sup_{\mathbf{w} \in \mathcal{S}_d} \left| \frac{1}{t} \sum_{i=1}^t |\mathbf{w}^T \mathbf{x}_i|^2 - 1 \right| \le \epsilon,$$

*and the observations $\mathbf{z}_i$ are normalized by $\ell_2$-norm, then for any $\mathbf{w}_1, \cdots, \mathbf{w}_d \in \mathcal{S}_p$, the following holds:*

$$\frac{(1-\epsilon)H(\mathbf{w}) - 2\sqrt{(1+\epsilon)H(\mathbf{w})}/s}{(1/s+1)^2} \le \frac{1}{t} \sum_{i=1}^t \sum_{j=1}^d (\mathbf{w}_j^T \mathbf{z}_i)^2 \le \frac{(1+\epsilon)H(\mathbf{w}) + 2\sqrt{(1+\epsilon)H(\mathbf{w})}/s + 1/s^2}{(1/s-1)^2},$$

*where $H(\mathbf{w}) = \sum_{j=1}^d \|\mathbf{w}_j^T A\|^2$ and $s$ is the signal noise ratio.*

*Proof.* Suppose the noise magnitude is $\|\mathbf{n}_i\|_2 = 1$ with out loss of generality. And thus the signal magnitude is $\|\mathbf{x}_i\|_2 = s$. Then we have:

$$
\frac{1}{t}\sum_{i=1}^{t}\sum_{j=1}^{d}\frac{|\mathbf{w}_j^T\mathbf{z}_i|^2}{\|\mathbf{z}_i\|_2^2}
$$

$$
= \frac{1}{t}\sum_{i=1}^{t}\sum_{j=1}^{d}\frac{|\mathbf{w}_j^T A\mathbf{x}_i + \mathbf{w}_j^T\mathbf{n}_i|^2}{\|A\mathbf{x}_i + \mathbf{n}_i\|_2^2}
$$

$$
\overset{(a)}{\leq} \frac{1}{t}\sum_{i=1}^{t}\sum_{j=1}^{d}\frac{|\mathbf{w}_j^T A\mathbf{x}_i + \mathbf{w}_j^T\mathbf{n}_i|^2}{(\|A\mathbf{x}_i\|_2 - \|\mathbf{n}_i\|_2)^2}
$$

$$
= \frac{1}{t}\sum_{i=1}^{t}\sum_{j=1}^{d}\frac{|\mathbf{w}_j^T A\mathbf{x}_i + \mathbf{w}_j^T\mathbf{n}_i|^2}{(s-1)^2}
$$

$$
= \frac{1}{(s-1)^2}\frac{1}{t}\left\{\sum_{i=1}^{t}\sum_{j=1}^{d}(\mathbf{w}_j^T A\mathbf{x}_i)^2 + 2\sum_{i=1}^{t}\sum_{j=1}^{d}(\mathbf{w}_j^T A\mathbf{x}_i)(\mathbf{w}_j^T\mathbf{n}_i) + \sum_{i=1}^{t}\sum_{j=1}^{d}(\mathbf{w}_j^T\mathbf{n}_i)^2\right\}
$$

$$
\overset{(b)}{\leq} \frac{1}{(s-1)^2}\sum_{j=1}^{d}\left\{\|\mathbf{w}_j^T A\|_2^2\sup_{\mathbf{v}\in\mathcal{S}_d}\frac{1}{t}\sum_{i=1}^{t}|\mathbf{v}^T\mathbf{x}_i|^2 + 2\sqrt{\frac{1}{t}\sum_{i=1}^{t}(\mathbf{w}_j^T A\mathbf{x}_i)^2}\sqrt{\frac{1}{t}\sum_{i=1}^{t}(\mathbf{w}_j^T\mathbf{n}_i)^2} + \frac{1}{t}\sum_{i=1}^{t}(\mathbf{w}_j^T\mathbf{n}_i)^2\right\}
$$

$$
\overset{(c)}{\leq} \frac{(1+\epsilon)\|\mathbf{w}^T A\|^2 s^2 + 2\|\mathbf{w}^T A\|s\sqrt{1+\epsilon} + 1}{(s-1)^2}.
$$

Here the inequality $(a)$ is from the triangle inequality. The inequality $(b)$ is from that

$$
\sum_{i=1}^{t}|\mathbf{w}_j^T A\mathbf{x}_i|^2 = \mathbf{w}_j^T A\left(\sum_{i=1}^{t}\mathbf{x}_i\mathbf{x}_i^T\right)A^T\mathbf{w}_j = \|\mathbf{w}_j^T A\|_2^2\left\{\frac{\mathbf{w}_j^T A}{\|\mathbf{w}_j^T A\|_2}\left(\sum_{i=1}^{t}\mathbf{x}_i\mathbf{x}_i^T\right)\frac{A^T\mathbf{w}_j}{\|\mathbf{w}_j^T A\|_2}\right\}
$$

$$
\leq \|\mathbf{w}_j^T A\|_2^2\sup_{\mathbf{v}\in\mathcal{S}_d}\frac{1}{t}\sum_{i=1}^{t}|\mathbf{v}^T\mathbf{x}_i|^2,
$$

for the first term and the inequality $(\sum_i a_ib_i)^2 \leq (\sum_i a_i^2)(\sum_i b_i^2)$ for the second term. And the inequality $(c)$ is from the definition of $H(\mathbf{w})$ and applying Lemma 1.

Similarly, we have

$$
\frac{1}{t}\sum_{i=1}^{t}\sum_{j=1}^{d}\frac{|\mathbf{w}_j^T\mathbf{z}_i|^2}{\|\mathbf{z}_i\|_2^2}
$$

$$
= \frac{1}{t}\sum_{i=1}^{t}\sum_{j=1}^{d}\frac{|\mathbf{w}_j^T A\mathbf{x}_i + \mathbf{w}_j^T\mathbf{n}_i|^2}{\|A\mathbf{x}_i + \mathbf{n}_i\|_2^2}
$$

$$
\geq \frac{1}{t}\sum_{i=1}^{t}\sum_{j=1}^{d}\frac{|\mathbf{w}_j^T A\mathbf{x}_i|^2 - 2|\mathbf{w}_j^T A\mathbf{x}_i||\mathbf{w}_j^T\mathbf{n}_i|}{\|A\mathbf{x}_i + \mathbf{n}_i\|_2^2}
$$

$$
\geq \frac{1}{t}\sum_{i=1}^{t}\sum_{j=1}^{d}\frac{|\mathbf{w}_j^T A\mathbf{x}_i|^2 - 2|\mathbf{w}_j^T A\mathbf{x}_i||\mathbf{w}_j^T\mathbf{n}_i|}{(\|A\mathbf{x}_i\|_2 + \|\mathbf{n}_i\|_2)^2}
$$

$$
= \frac{1}{t}\sum_{i=1}^{t}\sum_{j=1}^{d}\frac{(\mathbf{w}_j^T A\mathbf{x}_i)^2 - 2|\mathbf{w}_j^T A\mathbf{x}_i||\mathbf{w}_j^T\mathbf{n}_i|}{(s+1)^2}
$$

$$
\geq \frac{(1-\epsilon)H(\mathbf{w})s^2 - 2s\sqrt{(1+\epsilon)H(\mathbf{w})}}{(s+1)^2}.
$$

Combining the above two results, we complete the proof. $\qquad\square$

# 3 Proof of Lemma 4

**Lemma 4.** *If there exists $\epsilon$ such that*

$$\sup_{\mathbf{w}\in\mathcal{S}_d}\left|\frac{1}{t}\sum_{i=1}^{t}|\mathbf{w}^T\mathbf{x}_i|^2-1\right|\leq\epsilon,$$

*and the observations $\{\mathbf{z}'_i\}_{i=1}^m$ are sampled as in Algorithm 1, then for any $\mathbf{w}_1,\cdots,\mathbf{w}_d\in\mathcal{S}_p$, with large probability, the following holds:*

$$\frac{(1-\epsilon)H(\mathbf{w})-2\sqrt{(1+\epsilon)H(\mathbf{w})}/s}{(1/s+1)^2b/m}-\delta$$

$$\leq\quad\frac{1}{t}\sum_{i=1}^{t}\sum_{j=1}^{d}(\mathbf{w}_j^T\mathbf{z}'_i)^2\leq\frac{(1+\epsilon)H(\mathbf{w})+2\sqrt{(1+\epsilon)H(\mathbf{w})}/s+1/s^2}{(1/s-1)^2b/m}+\delta,$$

*where $H(\mathbf{w})\triangleq\sum_{j=1}^{d}\|\mathbf{w}_j^TA\|^2$, $s$ is the signal noise ratio and $m$ is the number of sampled observations in each batch.*

*Proof.* According to Lemma 2, we have

$$\frac{(1-\epsilon)H(\mathbf{w})-2\sqrt{(1+\epsilon)H(\mathbf{w})}/s}{(1/s+1)^2}\leq\frac{1}{t}\sum_{i=1}^{t}\sum_{j=1}^{d}(\mathbf{w}_j^T\mathbf{z}_i)^2\leq\frac{(1+\epsilon)H(\mathbf{w})+2\sqrt{(1+\epsilon)H(\mathbf{w})}/s+1/s^2}{(1/s-1)^2}.$$

Next we will show that the method of sampling without replacement given in Algorithm 1 provides an unbiased estimation of $\frac{1}{t}\sum_{i=1}^{t}\sum_{j=1}^{d}(\mathbf{w}_j^T\mathbf{z}_i)^2$. To see this, we define the random variables $X_i=|\mathbf{w}^T\mathbf{z}_i|^2$ and $Y_i=X_i/bX_i$ which is sampled from $X_i$ with probability $p_i=X_i$ and re-scaled by $bX_i$ as in Algorithm 1. Then

$$\mathbb{E}[Y_i]=\sum_{i=1}^{t}p_iY_i=\sum_{i=1}^{t}X_i\frac{X_i}{bX_i}=\sum_{i=1}^{t}\frac{X_i}{b}.$$

Thus,

$$\mathbb{E}\left[\sum_{i=1}^{m}Y_i\right]=\sum_{i=1}^{m}\mathbb{E}[Y_i]=\sum_{i=1}^{m}\sum_{i=1}^{t}\frac{X_i}{b}=\frac{m}{b}\sum_{i=1}^{t}X_i.$$

Namely,

$$\mathbb{E}\left[\sum_{i=1}^{m}\left|\mathbf{w}^T\mathbf{z}'_i\right|^2\right]=\frac{m}{b}\sum_{i=1}^{t}\left|\mathbf{w}^T\mathbf{z}_i\right|^2.$$

Thus, according to Lemma 3, we have

$$\left|\sum_{i=1}^{m}\left|\mathbf{w}^T\mathbf{z}'_i\right|^2-\frac{m}{b}\sum_{i=1}^{t}\left|\mathbf{w}^T\mathbf{z}_i\right|^2\right|=\left|\sum_{i=1}^{m}\left|\mathbf{w}^T\mathbf{z}'_i\right|^2-\mathbb{E}\left[\sum_{i=1}^{m}\left|\mathbf{w}^T\mathbf{z}'_i\right|^2\right]\right|<\delta,$$

with probability larger than $1-2\exp(-\delta^2/4m)$. Therefore,

$$\frac{m}{b}\sum_{i=1}^{t}\left|\mathbf{w}^T\mathbf{z}_i\right|^2-\delta\leq\sum_{i=1}^{m}\left|\mathbf{w}^T\mathbf{z}'_i\right|^2\leq\frac{m}{b}\sum_{i=1}^{t}\left|\mathbf{w}^T\mathbf{z}_i\right|^2+\delta.$$

Then applying Lemma 2 completes the proof. □

# 4 Proof of Lemma 5

**Lemma 5.** *For the current obtained principal components $\{\mathbf{w}_j^{(t-1)}\}_{j=1}^d$, the number of the accepted authentic samples $|\mathcal{Z}_t|$ and outliers $|\mathcal{O}_t|$ satisfy*

$$\left| \frac{|\mathcal{Z}_t|}{b} - \frac{1}{b} \sum_{i=1}^{(1-\lambda)b} \sum_{j=1}^{d} (\mathbf{w}_j^{(t-1)^T} \mathbf{z}_i)^2 \right| \leq \delta$$

*and*

$$\left| \frac{|\mathcal{O}_t|}{b} - \frac{1}{b} \sum_{i=1}^{\lambda b} \sum_{j=1}^{d} (\mathbf{w}_j^{(t-1)^T} \mathbf{o}_i)^2 \right| \leq \delta$$

*with probability at least $1 - e^{-2\delta^2 b}$. Here $\delta > 0$ is a small constant, $\lambda$ is the outlier fraction and $b$ is the size of the small batch.*

*Proof.* According to the Algorithm 1, the probability of accepting an authentic sample is

$$\Pr\left(\mathbf{z}_i \text{ is accepted}\right) = \sum_{j=1}^{d} \left(\mathbf{w}_j^{(t-1)^T} \mathbf{z}_i\right)^2.$$

Since there are in total $(1 - \lambda)b$ authentic samples, we have

$$\mathbb{E}|\mathcal{Z}_t| = \sum_{i=1}^{(1-\lambda)b} \sum_{j=1}^{d} \left(\mathbf{w}_j^{(t-1)^T} \mathbf{z}_i\right)^2.$$

By applying the Chernoff bound, we have

$$\Pr\left\{||\mathcal{Z}_t| - \mathbb{E}|\mathcal{Z}_t|| < \delta b\right\} \geq 1 - e^{-2\delta^2 b}.$$

Thus,

$$\Pr\left\{ \left| \frac{|\mathcal{Z}_t|}{b} - \frac{1}{b} \sum_{i=1}^{(1-\lambda)b} \sum_{j=1}^{d} \left(\mathbf{w}_j^{(t-1)^T} \mathbf{z}_i\right)^2 \right| \leq \delta \right\} \geq 1 - e^{-2\delta^2 b}.$$

Similarly, the expectation of the number of accepted outliers is

$$\mathbb{E}|\mathcal{O}_t| = \sum_{i=1}^{\lambda b} \sum_{j=1}^{d} \left(\mathbf{w}_j^{(t-1)^T} \mathbf{o}_i\right)^2.$$

And applying Chernoff bound again, we obtain

$$\Pr\left\{ \left| \frac{|\mathcal{O}_t|}{b} - \frac{1}{b} \sum_{i=1}^{\lambda b} \sum_{j=1}^{d} \left(\mathbf{w}_j^{(t-1)^T} \mathbf{o}_i\right)^2 \right| < \delta \right\} \geq 1 - e^{-2\delta^2 b}.$$

$\square$

# 5 Proof of Lemma 6

**Lemma 6.** *For an outlier $\mathbf{o}_i$, an arbitrary orthogonal basis $\{\mathbf{w}_j\}_{j=1}^d$ and the groundtruth basis $\{\overline{\mathbf{w}}_j\}_{j=1}^d$ which satisfy that $\sum_{j=1}^{d} \mathbf{w}_j^T \mathbf{o}_i \geq \sum_{j=1}^{d} \overline{\mathbf{w}}_j^T \mathbf{o}_i$ and $\sum_{j=1}^{d} \overline{\mathbf{w}}_j^T \mathbf{w}_j \geq \sum_{j=1}^{d} \overline{\mathbf{w}}_j^T \mathbf{o}_i$, the value of $\sum_{j=1}^{d} \mathbf{w}_j^T \mathbf{o}_i$ is a monotonically decreasing function of $\sum_{j=1}^{d} \overline{\mathbf{w}}_j^T \mathbf{w}_j$.*

*Proof.* For the basis $\{\overline{\mathbf{w}}_j\}_{j=1}^d$ spanning the groundtruth subspace, we can always rotate these basis and align them to the estimated basis $\{\mathbf{w}_j\}_{j=1}^d$ to make sure that $\mathbf{o}_i, \mathbf{w}_j$ and $\overline{\mathbf{w}}_j$ lie within the same plane. We also denote the aligned basis as $\{\overline{\mathbf{w}}_j\}_{j=1}^d$ without causing confusion. For the single basis pair, $\mathbf{w}_j$ and $\overline{\mathbf{w}}_j$, it can be verified that

$$\mathbf{w}_j^T \mathbf{o}_i = (\overline{\mathbf{w}}_j^T \mathbf{o}_i)(\overline{\mathbf{w}}_j^T \mathbf{w}_j) + \sqrt{1 - (\overline{\mathbf{w}}_j^T \mathbf{o}_i)^2}\sqrt{1 - (\overline{\mathbf{w}}_j^T \mathbf{w}_j)^2},$$

when the basis $\mathbf{w}_j$ satisfies the stated conditions. Thus we have

$$\sum_{j=1}^d \mathbf{w}_j^T \mathbf{o}_i = \sum_{j=1}^d (\overline{\mathbf{w}}_j^T \mathbf{o}_i)(\overline{\mathbf{w}}_j^T \mathbf{w}_j) + \sum_{j=1}^d \sqrt{1 - (\overline{\mathbf{w}}_j^T \mathbf{o}_i)^2}\sqrt{1 - (\overline{\mathbf{w}}_j^T \mathbf{w}_j)^2}.$$

It is easy to verify that when $\sum_{j=1}^d \overline{\mathbf{w}}_j^T \mathbf{w}_j \geq \sum_{j=1}^d \overline{\mathbf{w}}_j^T \mathbf{o}_i$, the function

$$f(\mathbf{w}_j^T \overline{\mathbf{w}}_j) = \sum_{j=1}^d (\overline{\mathbf{w}}_j^T \mathbf{o}_i)(\overline{\mathbf{w}}_j^T \mathbf{w}_j) + \sum_{j=1}^d \sqrt{1 - (\overline{\mathbf{w}}_j^T \mathbf{o}_i)^2}\sqrt{1 - (\overline{\mathbf{w}}_j^T \mathbf{w}_j)^2}$$

is a monotonically decreasing function w.r.t. $\sum_{j=1}^d \mathbf{w}_j^T \overline{\mathbf{w}}_j$ be seeing that the increase of any $|\mathbf{w}_j^T \mathbf{o}_i|$ will decrease value of the function. $\square$

## 6 Proof of Theorem 1

**Theorem 1** (Noisy Case Performance). *There exist constants $c_1', c_2'$ which depend on the signal noise ratio $s$ and $\epsilon_1, \epsilon_2 > 0$ which approximate zero when $s \to \infty$ or $b \to \infty$, such that if the outliers satisfies that $\sum_{j=1}^d |\overline{\mathbf{w}}_j^T \mathbf{o}_i|^2 \leq \Gamma_o$, the initial solution $\{\mathbf{w}^{(0)}\}_{j=1}^d$ in Algorithm 1 satisfies:*

$$\sum_{i=1}^{\lambda b} \sum_{j=1}^d \left| \mathbf{w}_j^{(0)T} \mathbf{o}_i \right|^2 \leq \frac{(1-\lambda)b(1-\epsilon^2)}{c_2'(1-\Gamma_o)} \left( \frac{1}{4}(c_1'(1-\epsilon) + \epsilon_1)^2 - \epsilon_2 \right),$$

*and*

$$H(\mathbf{w}^{(0)}) \geq \frac{1}{2}(c_1'(1-2\epsilon) + \epsilon_1) - \sqrt{\frac{(c_1'(1-\epsilon) + \epsilon_1)^2 - 4\epsilon_2}{4} - \frac{c_2' \sum_{i=1}^{\lambda b} \sum_{j=1}^d (\mathbf{w}_j^{(0)T} \mathbf{o}_i)^2 (1-\Gamma_o)}{(1-\lambda)b(1-\epsilon^2)}},$$

*then the performance of the solution from Algorithm 1 will be improved in each iteration, and eventually converges to:*

$$\lim_{t \to \infty} H_t \geq \frac{1}{2}(c_1'(1-2\epsilon) + \epsilon_1) + \sqrt{\frac{(c_1'(1-2\epsilon) + \epsilon_1)^2 - 4\epsilon_2}{4} - \frac{c_2' \sum_{i=1}^{\lambda b} \sum_{j=1}^d (\mathbf{w}_j^{(0)T} \mathbf{o}_i)^2 (1-\Gamma_o)}{(1-\lambda)b(1-\epsilon^2)}}.$$

*Here $\epsilon_1$ and $\epsilon_2$ decay as $\tilde{O}(d^{\frac{1}{2}} b^{-\frac{1}{2}} s^{-1})$, and $\epsilon$ decays as $\tilde{O}(d^{\frac{1}{2}} b^{-\frac{1}{2}})$. And $c_1' = (s+1)^2/(s-1)^2, c_2' = (1+1/s)^4$.*

*Proof of Theorem 1.* The sample covariance matrix at trial $t$ is calculated as:

$$C_t = \sum_{\mathbf{z}_i \in \mathcal{Z}_t} \mathbf{z}_i \mathbf{z}_i^T + \sum_{\mathbf{o}_i \in \mathcal{O}_t} \mathbf{o}_i \mathbf{o}_i^T.$$

And we have,

$$\sum_{j=1}^d \overline{\mathbf{w}}_j^T C_t \overline{\mathbf{w}}_j = \sum_{j=1}^d \sum_{\mathbf{z}_i \in \mathcal{Z}_t} (\overline{\mathbf{w}}_j^T \mathbf{z}_i)^2 + \sum_{j=1}^d \sum_{\mathbf{o}_i \in \mathcal{O}_t} (\overline{\mathbf{w}}_j^T \mathbf{o}_i)^2.$$

Thus for the PCA solution $\{\mathbf{w}_j^{(t)}\}_{j=1}^d$ on the current accepted data set $\mathcal{Y}_t = \mathcal{Z}_t \cup \mathcal{O}_t$, we have

$$\sum_{j=1}^d \sum_{\mathbf{z}_i \in \mathcal{Z}_t} \left( \mathbf{w}_j^{(t)T} \mathbf{z}_i \right)^2 + \sum_{j=1}^d \sum_{\mathbf{o}_i \in \mathcal{O}_t} \left( \mathbf{w}_j^{(t)T} \mathbf{o}_i \right)^2 \geq \sum_{j=1}^d \sum_{\mathbf{z}_i \in \mathcal{Z}_t} \left( \overline{\mathbf{w}}_j^T \mathbf{z}_i \right)^2 + \sum_{j=1}^d \sum_{\mathbf{o}_i \in \mathcal{O}_t} \left( \overline{\mathbf{w}}_j^T \mathbf{o}_i \right)^2,$$

$$(1)$$

where the inequality is from the fact that $\{\mathbf{w}_j^{(t)}\}_{j=1}^d$ are the leading eigenvectors of the covariance matrix $C_t$.

Note that all the data points are normalized by their $\ell_2$-norm, therefore $\sum_{j=1}^d \left(\mathbf{w}_j^{(t)^T}\mathbf{o}_i\right)^2 \leq 1$.

Thus we have $\sum_{j=1}^d \sum_{\mathbf{o}_i \in \mathcal{O}_t} \mathbf{w}_j^{(t)^T}\mathbf{o}_i^T\mathbf{o}_i\mathbf{w}_j^{(t)} \leq |\mathcal{O}_t|$. Substituting it to (1), we can obtain

$$\sum_{j=1}^d \sum_{\mathbf{z}_i \in \mathcal{Z}_t} \left(\mathbf{w}_j^{(t)^T}\mathbf{z}_i\right)^2 + |\mathcal{O}_t| \geq \sum_{j=1}^d \sum_{\mathbf{z}_i \in \mathcal{Z}_t} \left(\overline{\mathbf{w}}_j^T\mathbf{z}_i\right)^2 + \sum_{j=1}^d \sum_{\mathbf{o}_i \in \mathcal{O}_t} \left(\overline{\mathbf{w}}_j^T\mathbf{o}_i\right)^2 .$$

According to the definition of outliers, the outliers variance along the true PC directions is upper bounded, *i.e.*, $\sum_{j=1}^d \left(\overline{\mathbf{w}}_j^T\mathbf{o}_i\right)^2 \leq \Gamma_o$. Thus, we have

$$\frac{1}{|\mathcal{Z}_t|}\sum_{j=1}^d \sum_{\mathbf{z}_i \in \mathcal{Z}_t} \left(\mathbf{w}_j^{(t)^T}\mathbf{z}_i\right)^2 \geq \frac{1}{|\mathcal{Z}_t|}\sum_{j=1}^d \sum_{\mathbf{z}_i \in \mathcal{Z}_t} (\overline{\mathbf{w}}_j^T\mathbf{z}_i)^2 - \frac{|\mathcal{O}_t|}{|\mathcal{Z}_t|}(1 - \Gamma_o). \tag{2}$$

According to Lemma 4, we have followings hold with large probability $1 - 2\exp(-\delta^2/4m)$,

$$\frac{1}{|\mathcal{Z}_t|}\sum_{j=1}^d \sum_{\mathbf{z}_i \in \mathcal{Z}_t} \left(\mathbf{w}_j^{(t)^T}\mathbf{z}_i\right)^2 \leq \frac{(1+\epsilon)s^2 H^{(t)} + 2s\sqrt{(1+\epsilon)H^{(t)}} + 1}{(s-1)^2 b/m} + \delta, \tag{3}$$

and

$$\frac{1}{|\mathcal{Z}_t|}\sum_{j=1}^d \sum_{\mathbf{z}_i \in \mathcal{Z}_t} (\overline{\mathbf{w}}_j^T\mathbf{z}_i)^2 \geq \frac{(1-\epsilon)s^2 - 2s\sqrt{(1+\epsilon)}}{(s+1)^2 b/m} - \delta. \tag{4}$$

Here $m = |\mathcal{Z}^{(t)}|$, $H^{(t)} = H\left(\mathbf{w}_1^{(t)}, \ldots, \mathbf{w}_d^{(t)}\right)$ and we utilize the fact that $H(\overline{\mathbf{w}}_1, \ldots, \overline{\mathbf{w}}_d) = 1$. Substitute (3) and (4) to (2), we can obtain that

$$\frac{(1+\epsilon)H^{(t)} + 2\sqrt{(1+\epsilon)H^{(t)}}/s + 1/s^2}{(1/s - 1)^2} + 2\delta\frac{b}{|\mathcal{Z}^{(t)}|} \geq \frac{(1-\epsilon) - 2\sqrt{(1+\epsilon)}/s}{(1/s + 1)^2} - \frac{b|\mathcal{O}_t|}{|\mathcal{Z}_t|^2}(1 - \Gamma_o).$$

According to Lemma 5, we have, with a large probability,

$$\frac{|\mathcal{O}_t|}{|\mathcal{Z}_t|} = \frac{\sum_{j=1}^d \sum_{i=1}^{\lambda b} \left(\mathbf{w}_j^{(t-1)^T}\mathbf{w}_o\right)^2}{\sum_{j=1}^d \sum_{i=1}^{(1-\lambda)b} \left(\mathbf{w}_j^{(t-1)^T}\mathbf{z}_i\right)^2} \leq \frac{(1/s + 1)^2 \sum_{j=1}^d \sum_{i=1}^{\lambda b} \left(\mathbf{w}_j^{(t-1)^T}\mathbf{w}_o\right)^2}{(1-\lambda)b\left((1-\epsilon)H^{(t)} - 2\sqrt{(1+\epsilon)H^{(t)}}/s\right)},$$

and

$$\frac{|\mathcal{Z}_t|}{b} \geq (1-\lambda)\left((1-\epsilon)H^{(t)} - 2\sqrt{(1+\epsilon)H^{(t)}}/s\right).$$

Here the inequality is from Lemma 3.

Thus,

$$H^{(t)} \geq c_1 - \frac{c_2 \sum_{j=1}^d \sum_{i=1}^{\lambda b} \left(\mathbf{w}_j^{(t-1)^T}\mathbf{w}_o\right)^2 (1 - \Gamma_o)}{(1-\lambda)b((1-\epsilon)H^{(t-1)} - \epsilon')} - \bar{\epsilon} \tag{5}$$

where

$$c_1'(1 - 2\epsilon) \leq c_1 = \frac{(s-1)^2(1-\epsilon)}{(s+1)^2(1+\epsilon)} \leq c_1'(1-\epsilon),$$

$$c_2 = \frac{(1+1/s)^4}{1+\epsilon} = \frac{c_2'}{1+\epsilon},$$

$$\bar{\epsilon} = \frac{4(s^2+1)}{(s-1)^2 s\sqrt{1+\epsilon}} + \frac{1}{(1+\epsilon)s^2},$$

$$\epsilon' = 2\sqrt{(1+\epsilon)\bar{c}}/s.$$

Here, $c_1' = (s-1)^2/(s+1)^2, c_2' = (1+1/s)^4$.

In obtaining the above inequality (5), we utilize the fact that $H^{(t-1)} \leq 1$.

Based on the bound provided in (5), the result of Theorem 1 can be proved by induction. For the PC obtained from the first batch, $\{\mathbf{w}_j^{(1)}\}_{j=1}^d$, we have that

$$H^{(1)} \geq c_1 - \frac{c_2 \sum_{j=1}^d \sum_{i=1}^{\lambda b} \left(\mathbf{w}_j^{(0)T} \mathbf{o}_i\right)^2 (1-\Gamma_o)}{(1-\lambda)b((1-\epsilon)H^{(0)} - \epsilon')} - \bar{\epsilon}.$$

When the initial solution $\{\mathbf{w}_j^{(0)}\}_{j=1}^d$ satisfies the following conditions:

$$\sum_{i=1}^{\lambda b} \sum_{j=1}^d \left(\mathbf{w}_j^{(0)T} \mathbf{o}_i\right)^2 \leq \frac{(1-\lambda)b(1-\epsilon)}{c_2(1-\Gamma_o)} \left(\frac{1}{4}(c_1 - \epsilon_1)^2 - \epsilon_2\right) \overset{(a)}{\leq} \frac{(1-\lambda)b(1-\epsilon^2)}{c_2'(1-\Gamma_o)} \left(\frac{1}{4}(c_1'(1-\epsilon) + \epsilon_1)^2 - \epsilon_2\right),$$

(6)

and

$$H^{(0)} \geq \frac{1}{2}(c_1 - \epsilon_1) - \sqrt{\frac{1}{4}(c_1 - \epsilon_1)^2 - \epsilon_2 - \frac{c_2 \sum_{i=1}^{\lambda b} \sum_{j=1}^d \left(\mathbf{w}_j^{(0)T} \mathbf{o}_i\right)^2 (1-\Gamma_o)}{(1-\lambda)b(1-\epsilon)}},$$

(7)

$$\overset{(b)}{\geq} \frac{1}{2}(c_1'(1-2\epsilon) + \epsilon_1) - \sqrt{\frac{(c_1'(1-\epsilon) + \epsilon_1)^2 - 4\epsilon_2}{4} - \frac{c_2' \sum_{i=1}^{\lambda b} \sum_{j=1}^d (\mathbf{w}_j^{(0)T} \mathbf{o}_i)^2 (1-\Gamma_o)}{(1-\lambda)b(1-\epsilon^2)}}$$

where

$$\epsilon_1 = \bar{\epsilon} - \frac{\epsilon'}{1-\epsilon}, \text{ and } \epsilon_2 = \frac{\epsilon'(c_1 - \bar{\epsilon})}{1-\epsilon},$$

and the inequalities $(a)$ and $(b)$ are from the definitions of $c_1$ and $c_1'$, we can verify that

$$H^{(1)} \geq c_1'(1-2\epsilon) - \frac{c_2' \sum_{j=1}^d \sum_{i=1}^{\lambda b} \left(\mathbf{w}_j^{(0)T} \mathbf{w}_o\right)^2 (1-\Gamma_o)}{(1+\epsilon)(1-\lambda)b((1-\epsilon)H_0 - \epsilon')} - \bar{\epsilon} \geq H^{(0)}.$$

Since we are performing PCA on the authentic samples and outliers together, the obtained PCs $\{\mathbf{w}_j^{(0)}\}_{j=1}^d$ will span a subspace lying between a subset of the outliers and groundtruth subspace. For these outliers, the conditions stated in Lemma 6 are satisfied. Thus according to Lemma 6, we have

$$\sum_{i=1}^{\lambda b} \sum_{j=1}^d \left(\mathbf{w}_j^{(1)T} \mathbf{o}_i\right)^2 \leq \sum_{i=1}^{\lambda b} \sum_{j=1}^d \left(\mathbf{w}_j^{(0)T} \mathbf{o}_i\right)^2.$$

Similarly, suppose for the solution in $(t-1)$-th trial, we have $\sum_{i=1}^{\lambda b} \sum_{j=1}^d (\mathbf{w}_j^{(t-1)T} \mathbf{o}_i)^2 \leq \sum_{i=1}^{\lambda b} \sum_{j=1}^d (\mathbf{w}_j^{(0)T} \mathbf{o}_i)^2$. Thus,

$$H^{(t)} \geq c_1'(1-2\epsilon) - \frac{c_2' \sum_{i=1}^{\lambda b} \sum_{j=1}^d \left(\mathbf{w}_j^{(0)T} \mathbf{o}_i\right)^2 (1-\Gamma_o)}{(1+\epsilon)(1-\lambda)b((1-\epsilon)H^{(t-1)} - \epsilon')} - \bar{\epsilon}$$

And we can verify that when the initial solution satisfies the conditions (6) and (7), the performance of the new solution will be improved, namely

$$H^{(t)} \geq H^{(t-1)}.$$

And thus $\sum_{i=1}^{\lambda b} \sum_{j=1}^d \left(\mathbf{w}_j^{(t-1)T} \mathbf{o}_i\right)^2$ keeps decreasing according to Lemma 6.

Finally, by letting

$$c_1'(1-2\epsilon) - \frac{c_2' \sum_{i=1}^{\lambda b} \sum_{j=1}^{d} \left( \mathbf{w}_j^{(0)^T} \mathbf{o}_i \right)^2 (1-\Gamma_o)}{(1+\epsilon)(1-\lambda)b((1-\epsilon)H_{t-1} - \epsilon')} - \bar{\epsilon} = H^{(t)},$$

we can solve out that

$$H^{(t)} = \frac{1}{2}(c_1'(1-2\epsilon) - \epsilon_1) + \sqrt{\frac{(c_1'(1-2\epsilon) - \epsilon_1)^2}{4} - \epsilon_2 - \frac{c_2' \sum_{i=1}^{\lambda b} \sum_{j=1}^{d} \left( \mathbf{w}_j^{(0)^T} \mathbf{o}_i \right)^2 (1-\Gamma_o)}{(1-\lambda)b(1-\epsilon^2)}}.$$

Namely, the final performance will converge as above. $\qquad\square$

## 7  Simulations

Here we investigate the performance of proposed online RPCA under the cases where $d = 3$ and $d = 5$. The results are shown in the Figure 1 and Figure 2 respectively. We can see that in the cases where $d > 1$, the final performance will decrease a little bit. But along with more samples being revealed, the performance of online RPCA is steadily improved. This demonstrates the ability of our proposed online RPCA method to recover the underlying subspace even when the intrinsic dimension is large.

Figure 1: Performance comparison of online RPCA (blue line) with online PCA (red line). Here $s = 2, p = 100, T = 10,000, d = 3$. The outliers are distributed on a single direction.

Figure 2: Performance comparison of online RPCA (blue line) with online PCA (red line). Here $s = 2, p = 100, T = 10,000, d = 5$. The outliers are distributed on a single direction.