[Reviews · NeurIPS 2013]

Submitted by Assigned_Reviewer_6

This paper considers the theoretical properties of online updating of a "robust" PCA. Unlike traditional RPCA the proposed method does not learn the usual decomposition of the data into low-rank and sparse components but instead outliers are handled by accepting observations into the estimate of the covariance matrix proportional to their variance along the previously estimated PCA directions and scaling the accepted observations by this probability.

The proposed approach seems novel and certainly looks to be simpler than other approaches solving the same problem which are referred to in the paper. However it would have been nice to see a comparison of these methods.

The technical contribution of the paper seems sound and illuminating however the important question remains: how likely is it that the initial condition on H(w^0) is satisfied. This also seems to limit the usefulness of the method in real applications since in the experiments the authors initialise with PCA computed with the known outliers removed -- perhaps RPCA or similar could be used to get the initial estimate?

It also seems like in high dimensions, there might be issues due to high dimensionality/low sample size when computing the initial estimates.

There are a few other questions I am left with:
What happens when the concept changes i.e. when the underlying data generating process changes?

I have also noticed some possibly minor mistakes/typos.
In section 3.1 some random variable \mu is mentioned but it is never defined and seemingly not mentioned again. Is this a typo?

The sentence introducing \lambda is quite strange, it implies that \lambda has already been defined but in fact it has not.

In several places "affect" is mixed up with "effect".

Captions in Figs 2 and 3 are wrong. They should be number of batches.

It would have been instructive to compare the results with batch mode RPCA and also report on timings to get an idea if there is a significant speedup.
Summary: I think the work represents a nice contribution however some of the introductory technical exposition is quite sloppy and needs to be sorted out. The experiments which are presented show promising results but it does feel like they are incomplete.

Submitted by Assigned_Reviewer_7

This work presents an algorithm for computing principal components in an online setting when the data are corrupted by outliers (a subset of the data come from another, arbitrary distribution). The authors give a procedure for recovering the principal components and prove convergence in the limit. The paper is technically sound, clearly written, and addresses an important issue in online recovery of significant directions in high-dimensional data. I have only a few questions, one regarding the initial conditions, the other about notation.

First, it seems that a good starting point is required for the algorithm. Section 3.2 mentions the requirements of the initial condition, Theorem 1 states them explicitly, but the consequences of this are not discussed in the text. Furthermore, the simulations are initialized using uncorrupted samples. How would this work in practice? What are the consequences of chosing an initial set of vectors closer to the outliers than to the true principal components? This seems important enough to warrant a discussion.

Definition 1: is \Gamma_0 an upper bound on how far the outliers deviate from the PC subspace? Shouldn't this sum, and other instances, be a projection (normalized inner product)? I am also confused about the first equation in Section 4. I expect the E.V. of the *first* principal component w_1 to be 1, while the sum over all principal components will be the sum of the singular values of A.

Signal to noise s is not defined in the text.
Summary: This is a technically sound paper, with a mostly clear exposition (save for a couple items that confused this reviewer), and results on simulated data that support the claims (but seem to be based on a strong assumption regarding initial conditions).

Submitted by Assigned_Reviewer_8

The authors propose a robust PCA algorithm that is online and has some optimality properties with respect to outliers in the data. This is the first algorithm/analysis for online RPCA with theoretical guarantees. This is a good paper. The background is solid, the analysis is clear and reasonable. The two questions I have are
1) why the E.V. is used and not other distance metrics between subspaces, there are a variety and I suspect the results will hold very similarly for all these cases.
2) what about the dimension d, how does one select this

Also, the algorithm seems similar to follow the lazy leader type of algorithms.
Summary: This seems to be the first algorithm/analysis for online RPCA with theoretical guarantees, given outliers.
Author Feedback

Author rebuttal: We thank all reviewers for their constructive suggestions and insightful comments. The point-to-point replies are as follows.

To reviewer 6

Q1. It would be nice to see a comparison of other approaches referred to in the paper.

A1. Thanks for the suggestion. We have discussed the main limitations of related approaches in Sec.2. And we will provide the experimental comparison with [15] and [20] in the future work.

Q2. How likely the initial condition is satisfied, perhaps RPCA could be used to get the initial estimate?

A2. Yes, we agree with that. Initial estimate can be obtained by applying batch RPCA or HRPCA on a small subset of the data. These batch methods are able to provide initial estimate with performance guarantee, which may satisfy the initial condition. In real applications, we can first perform batch RPCA to get an initial estimate and then apply online RPCA to process the large scale data efficiently.

The motivation of this work is to handle the case where the samples outnumber the capacity of the computer (both in computation and in particular in storage). Due to the resource limitation, batch RPCA can only be performed on a subset of samples and cannot make full use of the whole collected data. In contrast, our method provides a way to break the bottleneck from computer capacity to make use of the entire data set, which is of use for the big-data regime.

Q3. It also seems like in high dimensions, there might be issues due to high dimensionality/low sample size when computing the initial estimates.

A3. We agree. However, as mentioned in the response to Q2, we can apply the batch HRPCA on a subset of the data to get the initial estimate. HRPCA is particularly designed for the high-dimensional regime (see Xu et al, TIT, 2013). It can provide satisfactory initial estimate.

Q4. What happens when the concept changes i.e. when the underlying data generating process changes?

A4. Online RPCA is able to dynamically adapt to the changing data generating process. If the process does not change too fast to follow, online RPCA can obtain dynamic and good PCs estimation. This is another advantage of online RPCA compared with static batch RPCA.

Q5. Some random variable \mu is mentioned but it is never defined and mentioned again.

A5. Sorry for causing the confusion. As mentioned in line 118, \mu denotes the distribution of signal x. We will give the explicit definition of \mu in the revision.

Q6. The sentence introducing \lambda is quite strange….

A6. Sorry for missing the formal definition. \lambda denotes the fraction of outliers in the samples. We will add this definition.

Q7. It would have been instructive to compare the results with batch RPCA and report on timings to see if there is a significant speedup.

A7. Thanks for the suggestion. In this paper, we are concerned more about the *memory cost* when dealing with large-scale dataset, as mentioned in A2. We will provide the performance of batch RPCA as reference, and report the time cost in the revision.

Q8. Captions in Fig 2 and 3 should be number of batches.

A8. Thanks for pointing out the typo. We will revise them.

To reviewer 7

Q1. The simulations are initialized using uncorrupted samples. How would this work in practice? What is the correspondence of choosing an initial set of vectors closer to the outliers than true PCs?

A1. The initial estimate can be obtained by applying a batch RPCA or HRPCA on a small subset of the data. These batch methods are able to provide initial estimate with performance guarantee, which may satisfy the required initial condition. Please refer to A2 to Reviewer 6 for discussion.

If the initial vectors closer to outliers, the result depends on which (outliers or true samples) contribute more to the covariance matrix, i.e. the initial condition. If the initial condition is satisfied, the final performance is still lower bounded. If not, we cannot obtain guarantee on the final performance. We will provide discussions on this point.

Q2. Is \Gamma_0 an upper bound on how far the outliers deviate from the PC subspace? Shouldn’t this sum be a projection (normalized inner product)?

A2. Indeed \Gamma_0 is the projection. Since we assume that Pcs estimation w and outliers o are all *normalized* (see Alg.1 step 1)-a) where all the samples are L2-normalized), we directly use inner product in the definition of \Gamma_0. We will clarify this point

Q3. Confused by the first equation in section 4.
A3. Sorry for the typo causing confusion. We meant to say that the nuclear norm of A is scaled to be 1. Thus the E.V. of optimal PCs is 1. We will clarify it.

Q4. Signal to noise s is not defined in the text.
A4. We will add the definition of SNR in the further version.

To reviewer 8

Q1. Why the E.V. is used and not other distance metrics? I suspect the results will hold very similarly.

A1. We agree with that for other distance metric, the results will similarly hold. Here, the reason for using E.V. metric is that E.V. is the most natural way to evaluate how good PCs are – note that in noiseless case PCA is directly maximizing it. Subspace angles may not be a good metric. For example, if there are two true PCs, one of them has larger EV than the other, then if the solutions deviate the correct one - deviating from the second PC should be considered better than deviating from the first. This can be captured via E.V., but not subspace angles.

Q2. How does one select the dimension d?
A2. It is generally pre-defined by the user of the algorithm based on certain estimation method or some prior knowledge, similar to implementing the standard PCA. When such knowledge is not available, trial and error on different d seems necessary.

Q3. The algorithm seems similar to follow the lazy leader type of algorithms.

A3. We will add a comment and reference concerning lazy leaders type of algorithms.